# Emergence of New SARS-CoV2 Omicron Variants after the Change of Surveillance and Control Strategy

**DOI:** 10.3390/microorganisms10101954

**Published:** 2022-09-30

**Authors:** José María González Alba, Zulema Pérez-Martínez, José A. Boga, Susana Rojo-Alba, Juan Gómez de Oña, Marta E. Alvarez-Argüelles, Garbriel Martín Rodríguez, Isabel Costales Gonzalez, Ismael Huerta González, Eliecer Coto, Santiago Melón García

**Affiliations:** 1Unit of Virology, Microbiology Department, Hospital Universitario Central de Asturias, 33011 Oviedo, Spain; 2Instituto de Investigación Sanitario del Principado de Asturias (ISPA), 33011 Oviedo, Spain; 3Genetic Department, Hospital Universitario Central de Asturias, 33011 Oviedo, Spain; 4Epidemiological Surveillance Department, 33011 Oviedo, Spain

**Keywords:** Omicron, new variants, surveillance, phylogenetic analysis

## Abstract

In January 2022, there was a global and rapid surge of the Omicron variant of SARS-CoV-2 related to more transmission. This coincided with an increase in the incidence in Asturias, a region where rapid diagnosis and containment measures had limited the circulation of variants. Methods: From January to June 2022, 34,591 variants were determined by the SNP method. From them, 445 were characterized by the WGS method and classified following pangolin program and phylogenic analysis. Results: The Omicron variant went from being detected in 2438 (78%) samples in the first week of January 2021 to 4074 (98%) in the third week, according to the SNP method. Using the WGS method, 159 BA.1 (35.7%), 256 BA.2 (57.6%), 1 BA.4 (0.2%) and 10 BA.5 (2.2%) Omicron variants were found. Phylogenetic analysis detected that three new sub-clades, BA.2,3.5, BA.2.56 and BF1, were circulating. Conclusions: The increase in the incidence of SARS-CoV2 caused the circulation of new emerging variants. Viral evolution calls for continuous genomic surveillance.

## 1. Introduction

On 26 November 2021, the World Health Organization (WHO) declared the Omicron variant (B.1.1.529) of Severe Acute Respiratory Syndrome Coronavirus 2 (SARS-CoV-2), which was initially discovered in South Africa, as a variant of concern (VOC). This Omicron variant is of particular concern as it is efficiently transmitted and has a short replication time [1]. In January 2022, there was a rapid increase in the proportion of BA.2 globally, a subvariant from Omicron B.1.1.529 [2,3]. A study in Denmark showed that BA.2 was associated with a higher rate of secondary attacks and increased susceptibility to infection in both unvaccinated and vaccinated individuals compared to BA.1 [4].

High vaccination coverage and increased testing diagnoses have contributed to the observed impact of the pandemic in the last wave that supports the transition to a different strategy that monitors and targets actions to the people and areas of greatest vulnerability. The infection control and surveillance as proposed by the Spanish Ministry of Health reduces the pressure on mild or asymptomatic cases and their contacts. Quarantines are not proposed for contacts, nor is screening in health centers generally considered necessary [5]. However, the relaxation of containment measures and Omicron characteristics implies accepting a certain level of transmission.

In a context of continuous emergence of new variants, their identification is essential for useful epidemiological monitoring and the implementation of measures to control the epidemic [6,7]. In this study, Omicron variants, including new variants, circulating in Asturias (a region in the north of Spain) at the beginning of 2022 were identified using both the single nucleotide polymorphism (SNP) and whole genome sequencing (WGS) methods.

## 2. Materials and Methods

### 2.1. Sample Collection

Between January to June 2022, 34,591 samples from nasopharyngeal swabs belonging to SARS-CoV2-infected patients were characterized by “in-house” the single nucleotide polymorphism (SNP) method. Of them, 445 samples were selected to be characterized by the whole genome sequencing (WGS) method. Data on age, sex, origin and pangolin lineage were collected (Appendix A).

### 2.2. SNP Methods

SARS-CoV-2 positive samples were amplified by three “in-house” RT-qPCRs developed to discriminate N501Y, L452R/Q and Y145H mutations. Amplifications were performed according to the protocol described by Sandoval-Torrientes et al. (2021) using sequence-specific forward and reverse oligonucleotide primers, along with TaqMan MGB variant specific probes, each with a different reporter dye at the 5′ end and a non-fluorescent quencher (NFQ) at the 3′ end [8].

### 2.3. WGS Method

Random samples with high viral load were selected, prioritizing those with inconclusive patterns in the SNP study. Selected SARS-CoV-2 positive samples were sequenced by using the Ion AmpliSeq SARS-CoV-2 research panel following the instructions set out in the manufacturer’s user guide. Libraries were prepared on the Ion Chef system as described in the user’s guide. Amplified samples were then sequenced, using Ion 540 chips (Thermo Fisher Scientific, Waltham, MA, USA), with the Ion S5 system (Thermo Fisher Scientific, Waltham, MA, USA), following the instructions set out in the manufacturer’s user guide. The obtained sequences were uploaded to GISAID (https://www.gisaid.org/ (accessed on 30 June 2022))

### 2.4. Classification/Characterization

The SARS-CoV-2 genomes available on the GISAID database to June 2022 were aligned using MAFFT (https://mafft.cbrc.jp/alignment/software/ (accessed on 30 January 2022)) and then manually curated using MEGA 7 (https://www.megasoftware.net/ (accessed on 16 June 2022)). Genomes covering more than 90% of the entire genome were included in the phylogenetic analysis. The best-fit nucleotide substitution model GTR + I was identified according to the Akaike information criterion using jModel-Test v2.1.10 (https://github.com/ddarriba/jmodeltest2 (accessed on 16 June 2022)). Phylogenetic trees were reconstructed by ML with FastTree (http://www.microbesonline.org/fasttree/(accessed on 16 June 2022)) for large trees or IQ-TREE (http://www.iqtree.org/ (accessed on 22 February 2022)). Bootstrap values were estimated using the SH test and ultrafast bootstrap. The bat Betacoronavirus sequence RaTG13 (EPI_ISL_402131) was used as an outgroup.

Dated phylogeny was reconstructed using Bayesian inference through a Markov chain Monte Carlo (MCMC) framework implemented in BEAST v1.10 (https://beast.community/ (accessed on 20 April 2022)) The days until the date of the most recent sequence were used as the sampling date. An uncorrelated relaxed clock model was employed to estimate the time to a most recent common ancestor (TMRCA). Bayesian Skyline analysis was used to infer how the population size is expected to change over time. MCMC chains were run for 200 million steps with sampling every 20,000 steps from the previous distribution. Convergence was evaluated by calculating the effective sample sizes of the parameters using Tracer v1.7.1 (https://beast.community/tracer (accessed on 20 April 2022)). Trees were summarized as maximum clade credibility trees using TreeAnnotator v1.8.4 after discarding the first 10% as burn-in, and then visualized in FigTree v1.4.4 (http://tree.bio.ed.ac.uk/software/figtree/(accessed on 15 November 2021)).

A phylogeographic analysis was performed in the BEAST program considering geographic locations as discrete states in a Bayesian statistical framework. An asymmetric substitution model and an uncorrelated relaxed molecular clock were applied to the Bayesian Stochastic Search Variable selection (BSSVS) method to identify the number of non-zero transitions (migrations) rates between states.

Using unusual SNPs in the circulating lineages in Asturias and monophyletic clades with more than five sequences and uploaded to GISAID, possible new Omicron lineages to study were defined. Lineage-specific mutations were obtained from analysis of WGS genomes against the Wuhan-Hu-1 reference genome (MN908947.3) and filtered for substitutions that comprise at least 99% of one lineage. The pattern of specific mutations of possible new lineages was searched for in the sequences obtained from GISAID in order to analyze their distribution around the world. New lineages or lineage refinements were suggested by submitting a topic at https://github.com/cov-lineages/pango-designation (accessed on 8 June 2022) [9].

## 3. Results

### 3.1. SNP Characterizations

Rapid characterization with the SNP method was obtained on 33,653 samples (1123 delta, 23,908 BA.1, 8569 BA.2 and 53 BA.4/BA.5). Figure 1 show weekly distribution of each variant according to Spike mutations.

### 3.2. WGS Characterizations

During the period studied, 159 BA.1 (35.7%), 256 BA.2 (57.6%), 1 BA.4 (0.2%) and 10 BA.5 (2.2%) Omicron variants were found. Delta variants were present in 19 (4.3%) samples. In a first analysis, within Omicron variants, 17 sub-variants were observed.

Table 1 compares the Omicron variants found both in Asturias and in the rest of the world in same period, and Figure 2 represents the phylogenetic relationship.

Further phylogenetic analysis showed that 63 samples were classified in three new clades of Omicron variants [9] (Table 2).

Twenty-six of them were BA.2.3 sequences, which gained the nsp2_T224I mutation generating the new sub-clade BA.2.3.5 (diverged in January). They were also found in England (8), USA (12), Spain (2), Denmark (1) and Scotland (1) (Figure 3A). Four sequences (EPI_ISL_11353524/EPI_ISL_11755009/EPI_ISL_12107391/EPI_ISL_1222106) came from a geriatric care center.

Another twenty-six BA.2 sequences gained the S_L452M mutation generating the new sub-clade BA.2.56 (diverged unknown), which was also found in Belgium (10), Germany (4), India (1), Japan (2), Netherlands (1), Singapore (1), Sweden (2), United Kingdom (68) and USA (2). The phylogeographic analysis estimates its origin in Belgium with a 99% probability (Figure 3B).

The remaining eleven BA.5 sequences gained the nsp16_V288F and nsp6_V289L mutations generating the new sub-clade BF.1, which was found in Denmark (4), England (7), Scotland (5) and Canada (1). The phylogeographic analysis does not allow the origin of the clade to be identified, but it is observed that the Asturian clade is different from the one that circulates in the rest of the world (Figure 3C).

In March, seven sequences of BA.1.17 (EPI_ISL_11353528, EPI_ISL_11353529, EPI_ISL_11353531, EPI_ISL_12107398, EPI_ISL_12107399, EPI_ISL_12107400, EPI_ISL_12107403) gained the nsp13_T115I mutation (Figure 2).

On the other hand, 17 BA.2 sequences (February to April) belonging to the same transmission clade were observed. They were similar to the others found in February in Galicia (Spain, *n* = 56), England (4) and USA (3) (Figure 2). The phylogeographic analysis estimates its origin in Galicia with a 92% probability.

## 4. Discussion

From the very beginning of SARS-CoV2 pandemic, Asturias reacted very quickly and designed methods to detect the virus and control the infection. High vaccination rates and the universal use of masks were considered important factors in slowing the spread of the virus that prevented many variants of the virus from appearing. In fact, as of August 2021, only 19 variants of the 40 found worldwide were in circulation [10].

At the beginning of December 2021, the Omicron variant appeared. Although the latest waves of the pandemic have been effectively contained, the appearance of the new Omicron variant poses future challenges [11].

At that moment, the Spanish Ministry of Health took the lead in adapting the general strategy followed until now, going on to monitor the impact of the disease on vulnerable people, hospitalizations and deaths, and monitor changes that may generate a modification in the favorable trends that are being observed at this time.

On the other hand, transmission from vaccinated infected persons is possible even if the disease is mild or asymptomatic, since vaccines reduce the probability of infection but are less effective in completely preventing virus replication in the mucosa of the upper respiratory tract of the vaccinated subject. Thus, variants with different characteristics of transmissibility, immune escape, and severity could emerge. We must, therefore, preserve the extraordinary surveillance and control structures developed during the pandemic and guarantee that they could be reactivated if necessary, either due to a worse evolution of the severity indicators or due to the appearance of new variants that condition this evolution [5].

At this time, the incidence began to increase in Asturias. At the end of December, the Omicron variant represented more than 75% of the cases. At the middle (10–16) of January 2022, the Omicron B.1.529 BA.1 variant accounted for 97% of the cases. That week the incidence of SARS-CoV2 virus was present in 684 cases in patients higher than 65 years old, the highest incidence registered [12].

This was also the beginning of the circulation of BA.2. After this moment, the BA.2 Omicron variant became the majority in the world, constituting half of the sequences uploaded to GISAID in the last month [2,3,13,14,15]. On average, BA.2 variants appeared only three months after being detected in the world, while BA.1 variants took five months.

The Omicron variant has a higher number of mutations compared to other variants and we will face more mutant variants in the future, which can result in significant changes in transmissibility, infectivity and pathogenicity [8,16,17].

First data on circulating variants were obtained with a fast SNP method that offers significant advantages in terms of cost, performance and efficient reporting of results and alerts about unexpected mutations. These sequences were analyzed by the WGS method, which allows the detection of new circulating variants [18].

The high incidence of the Omicron variant, as occurred in Asturias in January 2022, has led to many sub-lineages because virus replication is also increasing and mutations and recombinations are common. In fact, more than 100 sub-linages had been designed in May 2022 (www.gisaid.org (accessed on 30 May 2021)). Because there is a geographic distribution of variants, local circulation data are as important as national level data [13,15,19,20,21,22].

In Asturias, the clades and subclades of BA.2 found have an average of non-synonymousmutations that is higher than the number of mutations needed to characterize them using pangolin. Accordingly, phylogenetic analysis is needed.

No large outbreaks were identified among the clusters from the available data. Only a cluster of 17 BA.2 sequences were related to each other. This cluster was imported from the neighboring region (Galicia), which was circulating there in early February.

More interesting appears to be a BA.2.3 clade detected in early February that acquired the T44I mutation in the nsp2 gene and was subsequently designated as BA.2.3.5. Although this subclade is found in other countries, including Spain, the variant circulating in Asturias appears to be typical of the region. Phylogenetic analysis allowed a more precise classification since two of these sequences were initially classified as BA.2 due to incomplete sequencing. Belonging to this clade, one transmission event was found in an outbreak in a nursing home.

Another variant of interest is a BA.2 clade detected at the end of April (after the epidemic peak in April). This variant acquired the L452M mutation in the S gene and was subsequently designated BA.2.56. In addition to Asturias, BA.2.56 was found in other countries of the world (Germany, Belgium, India, Japan, Netherlands, Singapore, Sweden, United Kingdom, United States). Its country of origin could be Belgium, a country of Asturian emigration.

At the same time, at the end of April a new variant of omicron BA.5 initially identified in South Africa began to spread in Asturias. This variant gained the nsp16_V288F and nsp6_V289L mutations. No travel to South Africa was related and no similar variants circulated in Spain, so it could have been imported from one of the European countries to which Asturias is related. The phylogeographic analysis does not allow us to identify the origin of the clade, but it is observed that the Asturian clade is different from the one circulating in the rest of the world. Finally, this sub-variant was designated as BF.1 [9]. A sub-clade of BA1.17 that wins the mutation nsp13_T115I was circulating in March. This variant was only found in Asturias, and it could have been generated there, since incidence remains high. However, it was not proposed as a new sub-variant because it was not found anywhere else in the world and seems to be extinct.

Finally, a new sub-clade was found at the end of May from variant BA.5.1 and it is still circulating (data not shown, pending of confirmation) (issue 848 https:/github.com/cov-lineages/pango-designation/issues (accessed on 16 August 2021) [9]). This demonstrates the need for continued active surveillance to anticipate the potential effects of new virus variants.

On the other hand, rapid methods to determine possible variants (such as the SNP method that are consistent with the WGS data in terms of groups) could be very useful and a complement to the more specific but slower WGS method.

Unfortunately, there are little data on the epidemiological situation of patients. It is true that vaccination coverage in Asturias reaches 93.89%. The only data known are on patients who were institutionalized and where a specific variant was found.

## 5. Conclusions

The evolution of SARS-CoV2 to the Omicron variant, which appeared at the end of 2021, changed its earlier characteristics: apparently lower severity, but higher transmissibility and higher mutation rate. The high incidence in Asturias led to new sub-variants as in the rest of the world. Therefore, it is crucial to know if a new variant of the virus emerges to determine the re-infectivity rate, the transmissibility or the severity of the disease, especially after the removal of restrictive measures.

## Figures and Tables

**Figure 1 microorganisms-10-01954-f001:**
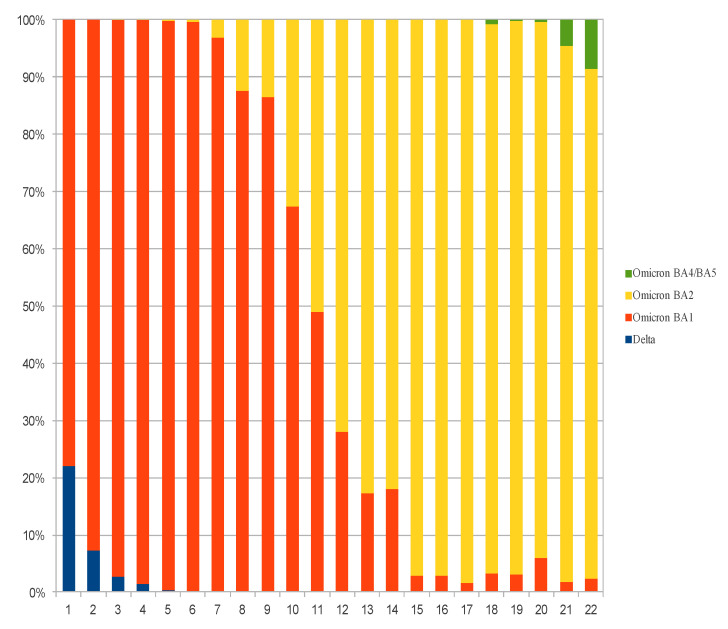
Weekly distribution of each variant in Asurias according Spike mutations.

**Figure 2 microorganisms-10-01954-f002:**
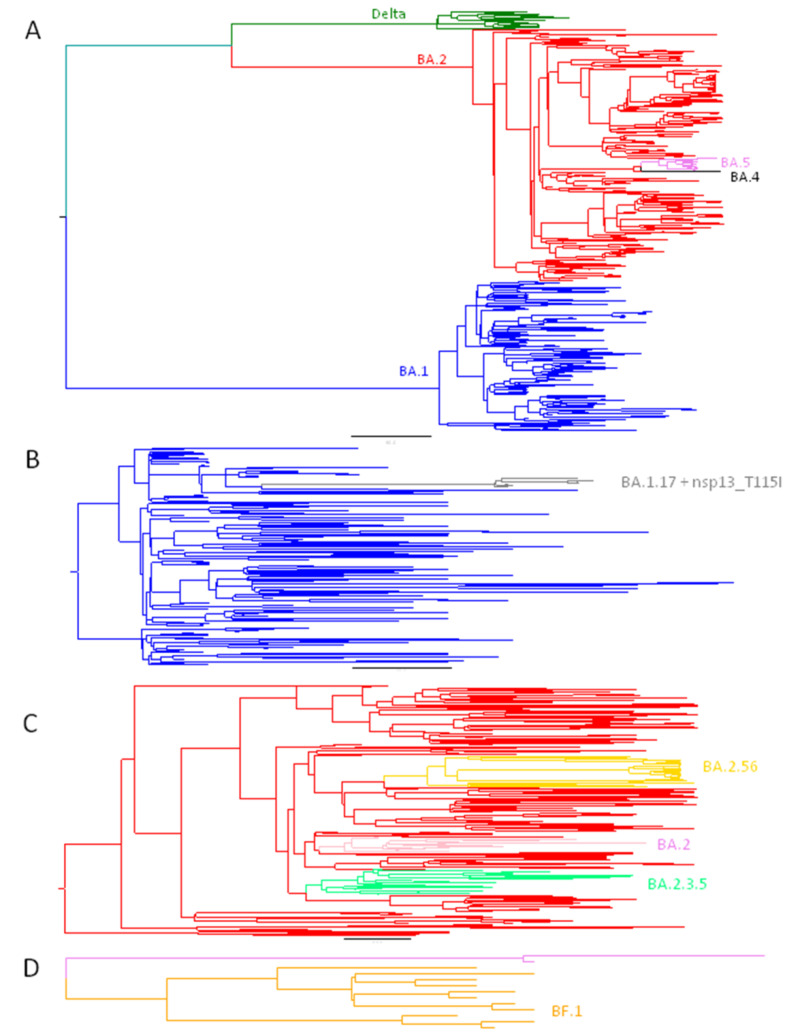
(**A**) Dated phylogeny of the 2022 GISAID Asturian sequences. Delta in green, BA.1 in blue, BA.2 in red, BA.4 in black and BA.5 in violet. (**B**) Dated phylogeny of BA.1 containing a monophyletic subclade of the BA.1.17 group found only in Asturias indicated in gray. (**C**) Dated phylogeny of BA.2 containing two monophyletic subclades found in Asturias and later defined as new groups, BA.2.56 in gold and BA.2.3.5 in spring green; and a monophyletic subclade that circulates in Asturias but has not evolved enough to be considered a new group in pink. (**D**) Dated phylogeny of BA.5 containing a monophyletic subclade found in Asturias later defined as a new group BF.1 in orange.

**Figure 3 microorganisms-10-01954-f003:**
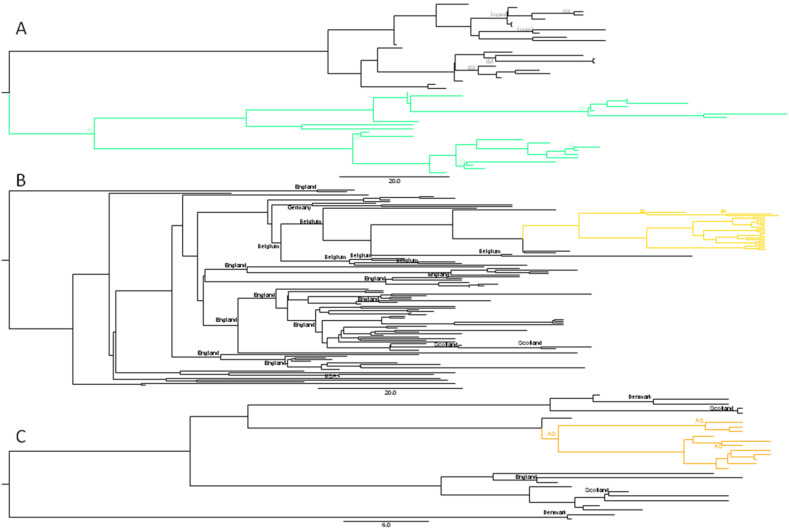
Phylogeographic analysis of the sequences deposited in GISAID belonging to the new Omicron lineages found in Asturias. The estimated country of origin with posterior probability > 90% is indicated in the nodes. AS is Asturias. (**A**) Lineage BA.2.3.5 with the Asturian sequences in spring green, (**B**) Lineage BA.2.56 with the Asturian sequences in gold and (**C**) Lineage BF.1 with the Asturian sequences in orange.

**Table 1 microorganisms-10-01954-t001:** Characteristics of Omicron variants BA.1 and BA.2 uploaded to GISAID in Asturias and the rest of the world between January to June 2022. Characteristic mutations are those defined by Pangolin.

Omicron Lineage	World			Asturias					
	N	Detection Date	Characteristic Mutations	n	Sex Female/Male	Age in Years	Detection Date	Arrival Time * (Days)	Mean of Mutations
BA.1	471,813	2021-04-14	43	25	11/14	51 (±28)	2022-01-08	269	45 (3.8)
BA.1.1	961,052	2021-05-15	46	33	16/17	57 (±28)	2022-01-08	238	44 (4.2)
BA.1.1.1	60,430	2021-11-16	45	42	20/22	54 (±27)	2022-01-08	53	44 (14.1)
BA.1.1.14	19,390	2021-12-03	45	5	2/3	46 (±36)	2022-02-19	78	46 (0.9)
BA.1.1.18	68,924	2021-08-23	45	1	1/-	38 (n.c.)	2022-03-01	190	43 (n.c.)
BA.1.17	61,198	2021-11-24	44	48	30/18	49 (±28)	2022-01-05	42	45 (2.9)
BA.1.17.2	187,655	2021-07-10	45	2	1/1	29 (±5)	2022-01-25	199	43 (0.8)
BA.1.18	54,260	2021-11-11	44	2	1/1	52 (±16)	2022-01-21	71	45 (2.1)
BA.1.8	1051	2021-12-08	47	1	1/0	2 (n.c.)	2022-03-02	84	42 (n.c.)
BA.2	1,013,860	2021-08-27	50	172	103/69	56 (±25)	2022-01-15	141	55 (4.3)
BA.2.12	10,355	2021-12-06	50	11	9/2	61 (±33)	2022-02-23	79	57 (2.1)
BA.2.18	4136	2022-01-17	49	3	1/2	67 (±19)	2022-05-11	114	56 (0.6)
BA.2.23	16,040	2021-12-09	49	1	-/1	43 (n.c.)	2022-04-26	138	54 (n.c.)
BA.2.3	73,493	2021-12-02	52	40	23/17	53 (±30)	2022-02-02	62	57 (2.2)
BA.2.9	156,403	2021-12-17	51	29	18	60 (±23)	2022-02-16	61	55 (2.6)
BA.4	2520	2022-01-10	52	1	1	56 (n.c.)	2022-05-13	123	64 (n.c.)
BA.5	2163	2022-01-05	50	10	6	54 (±18)	2022-04-25	110	58 (0.7)

* The estimated time until the most recent common ancestor of each group, which would be an estimate of its arrival in Asturias.

**Table 2 microorganisms-10-01954-t002:** New Omicron variants circulating in Asturias according pangolin program.

New Classification	Old Classification	Characteristic Mutations	*n*	Sex Female/Male	Age in Year	Detection Date	Mean of Mutations
BA.2.56	BA.2/BA.2.3	BA.2 + gene_S: L452M	26	12/14	69 (±14)	2022-04-24	56 (1.0)
BA.2.3.5	BA.2	BA.2.3 + orf1a_T224I	26	14/12	52 (±27)	2022-02-02	56 (2.4)
BF.1	BA.2/BA.5	BA.5 + nsp16_V288F, nsp6_V289L.	11	8/3	57 (±18)	2022-04-25	58 (0.5)

## Data Availability

Data availability in GISAID (https://www.gisaid.org/ (accessed on 30 June 2022).

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
