# Peer review of "Emergence of New SARS-CoV2 Omicron Variants after the Change of Surveillance and Control Strategy"

_microorganisms, 2022, doi:10.3390/microorganisms10101954_

Round 1

Reviewer 1 Report

The emergence of SARS-CoV-2 omicron subvariant surprised almost whole scientific world and together with COVID-19 vaccination led to some changes in the antiviral strategies in particular countries/regions. Gonzalez Alba and colleagues in their manuscript described the emergence of SARS-CoV-2 omicron variant at the beginning of 2022. Moreover, their data showed how omicron subvariant dominated the whole population just within few weeks. The manuscript is well-written and the data are quite clear, but I have some comments, please see below.

MAJOR COMMENTS

1.       According to the title, authors analysed the emergence of new SARS-CoV-2 subvariant after the change of surveillance and control strategy in Asturias. Unfortunately, I did not find any information how the anti-COVID-19 strategy was modified and how could this affect the emergence of new variant.

2.       Authors wrote that they analysed 445 out of 34591 samples with Next Generation Sequencing. How the samples for NGS were selected? What were the criteria?

3.       Table 1 and Figure 1 showed exactly the same data. Please decide whether you prefer to present them as table or as figure.

MINOR COMMENTS

-          Table 2 and 3 – what does the number in brackets in the ‘Age in years’ column mean?

-          Table 2 – what do you understand by ‘Arrival time’?

-          Table 2 and 3 – please provide the SD for the ‘Mean of mutations’

-          Figure 2 – it’s really hard to read it, please enlarge

-          Page 7 Line 165 – what do you understand by ‘unexpected sequences’? The SNP analysis cannot give unexpected sequences!

Author Response

Thank you very much for the reviewers' comments and suggestions. We have tried to follow your indications. Specifically:

MAJOR COMMENTS

  1. According to the title, authors analysed the emergence of new SARS-CoV-2 subvariant after the change of surveillance and control strategy in Asturias. Unfortunately, I did not find any information how the anti-COVID-19 strategy was modified and how could this affect the emergence of new variant.

The strategy change is defined in the document accessed with reference [5]. A different strategy aimed at people and areas of greater vulnerability, which implies accepting a certain level of transmission

https://www.sanidad.gob.es/profesionales/saludPublica/ccayes/alertasActual/nCov/documentos/Nueva_estrategia_vigilancia_y_control.pdf

  1. Authors wrote that they analysed 445 out of 34591 samples with Next Generation Sequencing. How the samples for NGS were selected? What were the criteria?

The number of sequences for NGS is related to our economic capacity to do massive sequencing, they were randomly chosen among samples with high viral load, but including all samples not typed or with unexpected mutations in the S gene by Sanger

Added phrase in WGS method : Random samples with high viral load were selected, prioritizing those with inconclusive patterns in the SNP study.

  1. Table 1 and Figure 1 showed exactly the same data. Please decide whether you prefer to present them as table or as figure.

We decided to leave the figure and to add the numbers of samples studied in results.

MINOR COMMENTS

-          Table 2 and 3 – what does the number in brackets in the ‘Age in years’ column mean?

This number means the standard deviation.  We put in the specific column of two tables "Age in years (SD)".

-          Table 2 – what do you understand by ‘Arrival time’?

The estimated time, with the available data using BEAST software, until the most recent common ancestor of each group, which would be an estimate of its arrival in Asturias .

Added phrase in Table:The estimated time until the most recent common ancestor of each group, which would be an estimate of its arrival in Asturias

-          Table 2 and 3 – please provide the SD for the ‘Mean of mutations’

The standard deviation added to the tables.

-          Figure 2 – it’s really hard to read it, please enlarge

Added in Figure:Figure 2. a) Dated phylogeny of the 2022 GISAID Asturian sequences . The groups found in Asturias are indicated with a different color. b) Dated phylogeny of BA.1 GISAID Asturian sequences. A monophyletic subclade of the BA .1.17 group found only in Asturias is indicated in red. c) Dated phylogeny of BA.2 GISAID Asturian sequences. Two monophyletic subclades of the BA .2 group found in Asturias later defined as new groups are indicated in color. A monophyletic subclade of BA.2 that circulates in Asturias but has not evolved enough to be considered a new group is also colored. d) Dated phylogeny of BA.5 GISAID Asturian sequences with colour. A monophyletic subclade of the BA .5 group found in Asturias later defined as a new group is indicated in color

-          Page 7 Line 165 – what do you understand by ‘unexpected sequences’? The SNP analysis cannot give unexpected sequences!

The mutations of the sequences are the unexpected ones, not the sequences;  “unexpected” was eliminated from the sentence.

Reviewer 2 Report

In their manuscript entitled "Emergence of new SARS-Cov2 omicron variants after the change of surveillance and control strategy", the authors monitored the circulation of SARS-CoV-2 variants in Asturias between Jan to Jun 2022 and they indentified 3 new Omicron variants circulating in this area. Their findings reiterate that continuous suivelliance is crucial for adopting proper countermeasures. Few concerns need to be addressed.

1, A total of 34591 samples were characterized by an in-house PCR method, of which 445 samples were sequenced. It is unclear whether the WGS data are in consistence with the SNP results.

2, Among the three Omicron variants, BA.2.56 is thought to be related to a variant identified in Belgium. While the other two are speculated to be locally cirulating variants. Did the authors ever try to track how these new variants occured? Was the infected individual unvaccined,  immune compromised or repeatedly infected?

3, Spelling errors, such as "Line 43, Gegome", should be corrected.

Author Response

Thank you very much for the reviewers' comments and suggestions. We have tried to follow your indications. Specifically:

In their manuscript entitled "Emergence of new SARS-Cov2 omicron variants after the change of surveillance and control strategy", the authors monitored the circulation of SARS-CoV-2 variants in Asturias between Jan to Jun 2022 and they indentified 3 new Omicron variants circulating in this area. Their findings reiterate that continuous suivelliance is crucial for adopting proper countermeasures. Few concerns need to be addressed.

1, A total of 34591 samples were characterized by an in-house PCR method, of which 445 samples were sequenced. It is unclear whether the WGS data are in consistence with the SNP results.

The NGS data are consistent with the SNP data in terms of groups (BA.1/BA.2/BA.3/BA.4/BA.5)

Added phrase in page 7 such as the SNP method that are consistent with the WGS data in terms of groups

2, Among the three Omicron variants, BA.2.56 is thought to be related to a variant identified in Belgium. While the other two are speculated to be locally cirulating variants. Did the authors ever try to track how these new variants occured? Was the infected individual unvaccined,  immune compromised or repeatedly infected?

Added phrase in page 7  Unfortunately, there is little data on the epidemiological situation of patients. It is true that vaccination coverage in Asturias reaches 93.89%. The only data known are on patients who were institutionalized and where a specific variant was found, as described in the manuscript.

3, Spelling errors, such as "Line 43, Gegome", should be corrected.

Done

Round 2

Reviewer 1 Report

I would like to thank the authors for their responses to my questions and suggestions. Unfortunately, in my opinion some points remain uncomplete. Please see below.

MAJOR COMMENTS

1.       I would strongly suggest to include in the manuscript changes of surveillance and control strategy in Asturias, which leads to the emergence of new SARS-CoV-2 genetic variant. The link, which you provided, leads to the document in Spanish, not understandable for non-Spanish speaking readers. In addition, the above-mentioned changes should be discussed in details in Discussion.

2.       The Figure 2 is completely messed up. It is impossible to see anything on this phylogenetic trees. Please rearrange and properly describe the phylogenetic branches. In the Legend please include the colours of described group.

3.       The Figure 3B and 3C are also not clear. You cannot understand what is compared if you do not provide some explanation and name at least some branches. At this moment the presented phylogenetic tree may compare different SARS-CoV-2 genetic variants, as well as Coronaviridae family in general.

MINOR COMMENTS

a.       Please do not use the SD value in the brackets in the Tables. Please use ± symbol.

Author Response

Thanks again for your suggestion. We have introduced a few sentences to improve the manuscript, as you suggest

MAJOR COMMENTS

  1. I would strongly suggest to include in the manuscript changes of surveillance and control strategy in Asturias, which leads to the emergence of new SARS-CoV-2 genetic variant. The link, which you provided, leads to the document in Spanish, not understandable for non-Spanish speaking readers. In addition, the above-mentioned changes should be discussed in details in Discussion.

Added on page 1 : The infection control and surveillance proposed by the Spanish Ministry of Health reduces the pressure on mild or asymptomatic cases and their contacts, quarantines for contacts are not proposed, nor is screening in health centers generally considered necessary [5]. However, the relaxation of containment measures and Omicron characeteristics implies accepting a certain level of transmission.

Added on page 6 : At that moment, the Spanish Ministry of Health leads to adapt the the general strategy followed until now, going on to monitor the impact of the disease on vulnerable people, hospitalizations and deaths, and monitor changes that may generate a modification in the favorable trends that are being observed at this time.lt

On the other hand, transmission from vaccinated infected persons is possible even if the disease is mild or asymptomatic since vaccines reduce the probability of infection, but are less effective in completely preventing virus replication in the mucosa of the upper respiratory tract of the vaccinated subject.  Then variants with different characteristics of transmissibility, immune escape, and severity could emerge. Should be therefore preserve the extraordinary surveillance and control structures developed during the pandemic and guarantee that they could be reactivated if necessary, either due to a worse evolution of the severity indicators or due to the appearance of new variants that condition this evolution [5].

  1. The Figure 2 is completely messed up. It is impossible to see anything on this phylogenetic trees. Please rearrange and properly describe the phylogenetic branches. In the Legend please include the colours of described group.

 Figure 2 changed

Figure 2. A)Dated phylogeny of the 2022 GISAID Asturian sequences . Delta in green, BA.1 in blue, BA.2 in red, BA.4 in black and BA.5 in violet. B ) Dated phylogeny of BA.1 containing a monophyletic subclade of the BA .1.17 group found only in Asturias indicated in grey. C) Dated phylogeny of BA.2 containing :two monophyletic subclades found in Asturias and later defined as new groups, BA.2.56 in gold and BA.2.3.5 in spring green; and  a monophyletic subclade  that circulates in Asturias but has not evolved enough to be considered a new group in pink. D) Dated phylogeny of BA.5 containing a monophyletic subclade  found in Asturias later defined as a new group BF.1 in orange.

  1. The Figure 3B and 3C are also not clear. You cannot understand what is compared if you do not provide some explanation and name at least some branches. At this moment the presented phylogenetic tree may compare different SARS-CoV-2 genetic variants, as well as Coronaviridae family in general.

 Figure 3 changed

Figure 3. Phylogeographic analysis of the sequences deposited in GISAID belonging to the new omicron lineages found in Asturias. The estimated country of origin with posterior probability > 90% is indicated in the nodes. AS is Asturias. A) Lineage BA.2.3.5 with the asturian sequences in spring green , B) Lineage BA.2.56 with the asturian sequences in gold and C) Lineage BF.1 with the asturian sequences in orange

MINOR COMMENTS

  1. Please do not use the SD value in the brackets in the Tables. Please use ±

Done